# Ultrafast Deep-Ultraviolet Laser-Induced Voltage Response of Pyrite

**DOI:** 10.3390/mi12121555

**Published:** 2021-12-13

**Authors:** Xuecong Liu, Yudong Li, Haoqiang Wu, Yawen Yu, Honglei Zhan, Xinyang Miao, Kun Zhao

**Affiliations:** 1College of Information Science and Engineering, China University of Petroleum, Beijing 102249, China; 2019215828@student.cup.edu.cn; 2Key Laboratory of Quantum Exploration Material and Technology, China University of Petroleum, Beijing 102249, China; 2020211141@student.cup.edu.cn (H.W.); 2020211142@student.cup.edu.cn (Y.Y.); zhanhl@cup.edu.cn (H.Z.); 3Beijing Key Laboratory of Optical Detection Technology for Oil and Gas, China University of Petroleum, Beijing 102249, China; 2020211127@student.cup.edu.cn; 4College of New Energy and Materials, China University of Petroleum, Beijing 102249, China; 5Key Laboratory of Oil and Gas Terahertz Spectroscopy and Photoelectric Detection, Petroleum and Chemical Industry Federation, China University of Petroleum, Beijing 102249, China

**Keywords:** pyrite, laser-induced voltage, ultrafast detection

## Abstract

Ultrafast, high-sensitivity deep-ultraviolet (UV) photodetectors are crucial for practical applications, including optical communication, ozone layer monitoring, flame detection, etc. However, fast-response UV photodetectors based on traditional materials suffer from issues of expensive production processes. Here, we focused on pyrite with simultaneously cheap production processes and ultrafast response speed. Nanoseconds photovoltaic response was observed under UV pulsed laser irradiation without an applied bias at room temperature. In addition, the response time of the laser-induced voltage (LIV) signals was ~20 ns, which was the same as the UV laser pulse width. The maximum value of the responsivity is 0.52 V/mJ and the minimum value of detectivity was about to ~1.4 × 10^13^ Jones. When there exists nonuniform illumination, a process of diffusion occurs by which the carriers migrate from the region of high concentration toward the region of low concentration. The response speed is limited by a factor of the diffusion of the carriers. With an increment in laser energy, the response speed of LIV is greatly improved. The high response speed combined with low-cost fabrication makes these UV photodetectors highly attractive for applications in ultrafast detection.

## 1. Introduction

Pyrite is an earth-abundant mineral semiconductor, which contains mainly iron, oxygen, sulphur, and trace elements such as Pb, Au and Mo [1]. Pyrite is a low-cost candidate material for photocatalysis, photovoltaic devices, photoelectrochemical applications, and energy storage batteries due to its unique properties such as high absorption coefficient (α > 105 cm^−1^ for hν > 1.3 eV), narrow energy band gap (∼0.95 eV), and high electron mobility (230 cm^2^/Vs) [2,3,4]. The Schottky junction type pyrite photovoltaic devices exhibit high quantum efficiency and photocurrent [5]. The theoretical limit of the power conversion efficiency of pyrite is better than that of CuInSe_2_ and Si-based photovoltaic devices [6]. Therefore, pyrite is of significant interest due to its low cost and high abundance [7].

With the irradiation of a laser, the electron–hole pairs are immediately separated by the built-in electric field [8]. Once the charge carriers are dissociated, the liberated electrons and holes are collected by the electrodes to deliver the electrical signal to the external circuit, with the voltage generated being denoted as laser-induced voltage (LIV) [9]. LIV response is an exciton phenomenon in which the laser energy exceeds the energy band gap of the sample. It is not only sensitive to the interior composition and structures of the sample, but also reflects variations in laser characteristics [10,11]. Recent studies have shown that LIV response can be employed to assess the structural characteristics of many materials. The different bedding directions of shale can be characterised by LIV responses [12]. Simultaneously, LIV can be used to evaluate oil yield in oil shale [13,14].

In recent decades, the rapid advance in semiconductor optoelectronic devices has transformed our world. Attempts have been made to study wide-band-gap semiconductor materials such as ZnO, GaN, TiO_2_ and SiC, which have been used in the fabrication of ultraviolet (UV) photodetectors [15,16,17,18,19]. With the development of optoelectronic integration technology, there is a high requirement for photodetectors with a self-sufficient mode of operation and cheap raw materials favouring the photoelectric effect. The ZnO, GaN, TiO_2_ and SiC-based UV photodetectors were found to exhibit issues regarding high cost, crystalline quality, and complicated fabrication techniques [20,21,22]. Meanwhile, pyrite has good potential for photoelectric response due to its characteristics of high natural abundance and nontoxic elements [2]. Since the absorption of pyrite in UV light (e.g., at 248 nm) is about four times that of visible light (e.g., at 550 nm), photodetectors based on pyrite are more sensitive to UV light than visible light [23].

As a natural sulphide mineral, the electronic structure of pyrite can be understood in terms of molecular orbital (MO) theory, which considers the formation of molecular orbitals between two atoms as the combination of their atomic orbitals [24]. For the FeS2, bonding and anti-bonding molecular orbitals are formed by the hybridised sp^3^ orbitals of the overlapped sulphur atoms and the d^2^sp^3^ iron atoms [25]. The MO model for pyrite is shown in Figure 1. In such a model, the two electrons in each sulphur hybrid sp^3^ orbital completely fill the σ molecular orbitals, while the non-bonding σ* and eg* orbitals in transition metal ions are unoccupied [26]. Under UV laser illumination, the 3d electrons of the transition metal occupy non-bonding σ* and eg* orbitals due to the higher photon energy than the bandgap of pyrite. It is assumed that conduction occurs when electrons are excited into the conduction band formed from the eg* orbitals [27]. In this case, the photogenerated carriers can be excited throughout the band structure.

Here, the pyrite was selected as detection material for the engineering of fast-responding deep-UV laser photodetectors. When the laser irradiates the pyrite, photons are absorbed to excite electron−hole pairs in the irradiation area, and diffused to the external circuit through the electrodes. High-speed voltage response of ~20 ns was achieved in pyrite under UV pulsed laser irradiation without an external bias at room temperature. The responsivity and detectivity of pyrite are 0.52 V/mJ and ~1.4 × 10^13^ Jones, respectively. The voltage responses strongly depend on the laser energy on the sample. The present results suggest that the pyrite can be used for ultrafast and effective detection of high-energy deep-UV laser.

## 2. Materials and Methods

The pyrite in this experiment was selected from Shangbao in the south-central part of 56 China, Hunan Province. Component and structure information of the pyrite were examined with X-ray diffraction (XRD) and scanning electron microscopy (SEM). For the LIV measurement, the pyrite samples, which have a size of 7 × 4 mm^2^ with a thickness of 2 mm, were mechanically fabricated by a fine Diamond Cutter. Two 2 × 4 mm^2^ colloidal silver electrodes were separated on the polished surface by 2 mm. The sample was placed in an airtight holder with an adjustable diaphragm and connected with the digital oscilloscope with a bandwidth of 350 MHz and input impedance of 1 MΩ to monitor the pulse response. A KrF pulsed laser with a wavelength of 248 nm and a pulse width of 20 ns was employed to irradiate the pyrite. The on-sample energy E_in_ with an irradiation area of 3 × 4 mm^2^ was changed from 27.6 to 47.5 mJ by increasing the laser energy density from 2.3 to 3.96 mJ/mm^2^. In addition, the energy intensity near the centre was relatively high and decreased along the radial direction for the pulsed laser with a Gaussian energy distribution. E_in_ was also varied from 0.25 to 13.55 mJ by changing the laser spot area on the surface from 1 to 16 mm^2^, where the output fluence of the pulse laser of the setup was ~1 mJ/mm^2^. All measurements were performed at 25 °C. The schematic diagram of the measurement system is shown in Figure 2.

## 3. Results and Discussion

Figure 3 shows the morphological and structure information of the pyrite samples. Cubes, the common euhedral crystal type, are bright under a reflective microscope. The grains possess an irregular shape and are randomly oriented. Pyrite framboids are generally spherical aggregates of submicron-size pyrite crystals. In Figure 3a,b, crystals and pyrite framboids with the average size of 0.3 μm are generally diffusely distributed. As shown in Figure 3c–e, the pyrite’s surface structures are various. The pyrite surface in Figure 3f appears relatively smooth and clean, with some naturally occurring voids and pits. Needle bar granular calcium feldspar are occasionally visible on the pyrite surface. Impurity heterogeneity, structural complexity and small changes in composition can lead to variation in formation of states near the fermi level. Therefore, the band structure of the sample changes with the influence of impurities [28].

The XRD patterns of the pyrite sample are displayed in Figure 4a, showing the main diffraction peaks of (1 1 1), (2 0 0), (2 1 0), (2 1 1), (2 2 0), (3 1 1), (2 2 2), (2 3 0), and (3 2 1), which correspond well to the characteristic peaks of a pyrite structure with high crystallinity. The crystal structure of the pyrite is shown in Figure 4b, and it exhibits a similar structure to that of the Pa3 space group. Therein, the distribution of the Fe atoms and S_2_ molecules are highlighted as blue and yellow balls. Fe atoms are situated at the face-centred site of one sublattice, while the S_2_ molecules are situated on the other, giving an NaCl-like structure [29]. Six S atoms and Fe atom bonding axes are oriented along the (1 1 1) crystallographic axis direction. The pyrite shows a wide diffraction peak at around 28° assigned to silicate, and the position and crystallinity of the pyrite diffraction peak is basically retained [30,31], indicating the dominant mineral phases as being pyrite with small amounts of silicate.

A typical voltage transient of pyrite is presented in Figure 5 (blue curve). When the sample surface was irradiated at t_1_ = 0 s under the laser pulse with E_in_ = 47.5 mJ, the pulse signal rose rapidly till the maximum value (V_p_) of 0.0196V at t_2_, and then slowly decreased. The rise time of ~15.8 ns, with the rise time being defined by t_2_ − t_1_, indicates that the carriers transferred inside the pyrite could be quickly captured by the silver electrodes and then conducted to the measuring oscilloscope. The attenuation part displayed some periodic oscillations persisting for hundreds of nanoseconds, which may be due to the signal reflection arising from an impedance mismatch in the circuit [32]. When the load achieves good impedance matching, the slow attenuation of the signal shows the carrier recombination process [33]. We need to address the superposition of the declining and oscillating processes. The downward trend of the voltage pulse was extracted by the yellow curve, as shown in Figure 4, where the t_3_ and t_4_ were defined as the times when upward and downward signal responses reached 1/2 V_p_. Thus, the full width at half maximum (FWHM), FWHM = t_4_ − t_3_, of ~29.6 ns, was measured.

Pyrite has an appropriate energy bandgap of ~0.95 eV, causing it to have excellent electron mobility and environmental compatibility [28]. When the sample receive a pulsed laser, photoexcited carriers was detected by the silver electrodes. As shown in Figure 6a, the LIV signals strongly depended on the laser fluence, with V_p_ monotonically increasing with energy density from 2.3 to 3.96 mJ /mm^2^. Time-dependent LIV under selected irradiation areas from 1 to 16 mm^2^ is shown in Figure 6b. V_p_ was significantly improved from 0.77 to 10 mV when the E_in_ increased, owing to the change to a laser irradiation area. Responsivity (R) is a critical factor for evaluating a detector which is defined as the ratio of (V_p_ − V_dark_) and the incident optical energy intensity E_in_, where V_dark_ is the noise voltage in the dark. The responsivity is expressed by R = (V_p_ − V_dark_)/E_in_. The maximum R is about 0.52 V/mJ in our case. The specific detectivity (D*) is a key figure-of-merit for describing the smallest detectable light density by a photodetector. This can be calculated according to D* = R*A^1/2^/(2eV_dark_/R_0_)^1/2^, where R is the calculated responsivity, A is the active area (in mm^2^), e is the electron charge and R_0_ is 1 MΩ. The minimum value of D* calculated for the measurement system fabricated in this study is found to be ~1.4 × 10^13^ Jones. The results suggested the pyrite device, which shows that the pyrite device is highly responsive and very sensitive to UV laser radiation.

When the laser irradiates the pyrite, some photons are reflected with constant reflectivity. The other impinged photons create electron–hole pairs, resulting in a generation rate correlated with E_in_. Therefore, the excess carrier density will be proportional to the laser energy [34]. As shown in Figure 7a, V_p_ increased with the increasing E_in_ by selecting different irradiation areas or laser energy densities. An insight study was performed to further inspect the rise time and FWHM under different values of E_in_, which is the crucial factor for determining the quality of a photodetection device. The rise time and FWHM are plotted as a function of E_in_ in Figure 7b. In the presence of inhomogeneous illumination, a gradient of carrier concentration occurs, with carriers diffusing outwards and regrouping to drive the signal change. For this process, the recombination rate is proportional to electron and hole concentrations. Both the rise time and FWHM showed high values of 27.74 ns and 45.24 ns at E_in_ = 0.25 mJ and drop gradually to 15.83 ns and 29.59 ns at E_in_ = 47.5 mJ with the increase in E_in_. The present relationships revealed that the E_in_ played a great role in the voltage response. Meanwhile, the response time was much faster than the polycrystalline thin film (hundreds of µs) and bulk single crystal UV photodetectors (tens of ms).

The current transport in metal–semiconductor contacts is due mainly to majority carriers. Because the barrier height is determined primarily by the characteristics of the metal and the metal–semiconductor interface properties, the Fermi level (E_F_) of Ag immediately after contact is lower than that of pyrite, and there is a potential difference at the interface [35]. Figure 8 displays the energy-band diagram for the interface between Ag and pyrite. When laser irradiates at the surface of the pyrite, carriers are generated either by band-to-band transitions or by transitions involving forbidden-gap energy levels, resulting in a change in conductivity. The laser radiation is first absorbed by the electrons from the valence band in pyrite within the ultrashort pulse duration and subsequently transferred to the conduction band through electron phonon interactions over a characteristic time [32]. At the interface, the band movement balances the Fermi energy level and transfers the excited electrons from the pyrite conduction band to the Ag electrode conduction band. Electrons in the conduction band migrate to the external circuit and are detected by the oscilloscope. Illumination always tends to decrease the band bending [36]. As a result of this, the transportation of charge carriers becomes very easy. With an increment in E_in_, the concentration of electron–hole pairs increases, and more and more electrons are attracted to the electrodes, where they increase the value of Vp. On the other hand, under laser irradiation with different Ein, photogenerated electrons diffuse to the nearby electrode at different velocities due to their different mobility. In particular, driven by the concentration gradient, electronics could travel faster to the further electrodes, which greatly improve the LIV response speed. In this way, the performance of the device increases tremendously.

In this research, we aimed to discuss the photoelectric response characteristics of pyrite irradiated by a 248 nm pulsed laser. The LIV response was determined to be dependent on the laser energy on the sample, with the LIV signal increasing with the Ein and the response time being negatively correlated with Ein. A maximum responsivity of 0.52 V/mJ and a detectivity of ~1.4 × 10^13^ Jones were obtained without external bias. The rise time and FWHM of 15.83 ns and 29.59 ns ensured the rapid detectivity of the device. As a result, as a rich sulphide, pyrite can be a promising high-performance candidate for deep-UV laser photodetection.

## 4. Conclusions

In summary, a systematic study was performed to realise a highly sensitive ultrafast UV detector using pyrite, which exhibited a response speed of ~16 ns, a responsivity of 0.52 V/mJ and a detectivity of ~1.4 × 10^13^ Jones at the same time. When the device is illuminated with pulsed UV laser, electron–hole pairs are excited, and this cumulative process increases the amplitude and rate of photoelectric generation. The charge carrier generation and transportation in the fabricated device are governed by energy level diagrams. The LIV responses strongly depend on the laser energy on the sample. Therefore, high-sensitivity ultra-fast UV detector can be developed by using the characteristics of low cost and low energy consumption of natural pyrite.

## Figures and Tables

**Figure 1 micromachines-12-01555-f001:**
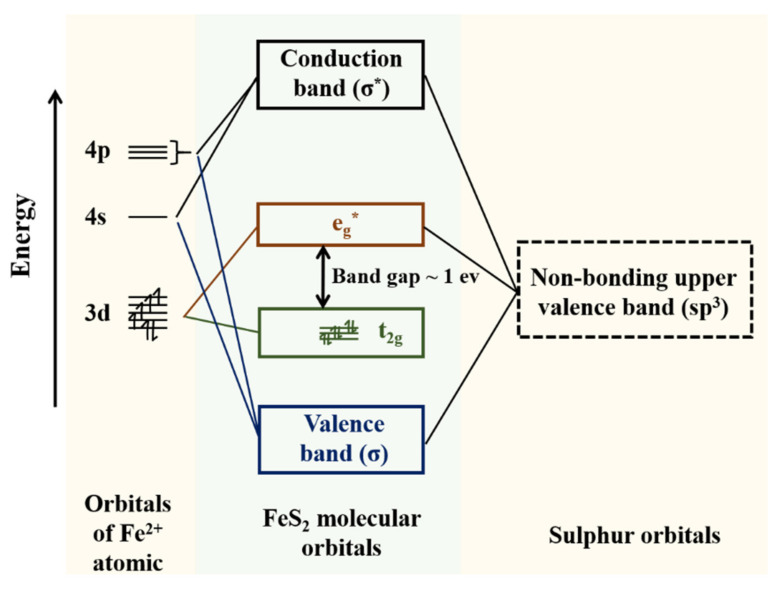
Molecular orbital energy level diagram for pyrite.

**Figure 2 micromachines-12-01555-f002:**
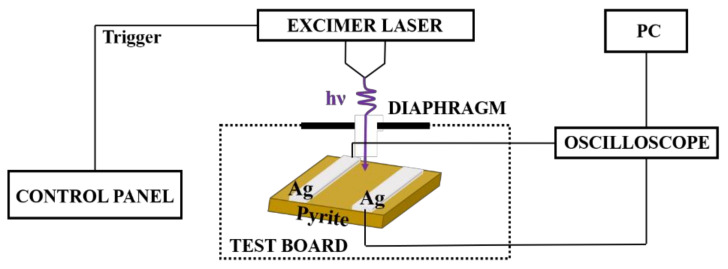
The schematic diagram of the measurement system.

**Figure 3 micromachines-12-01555-f003:**
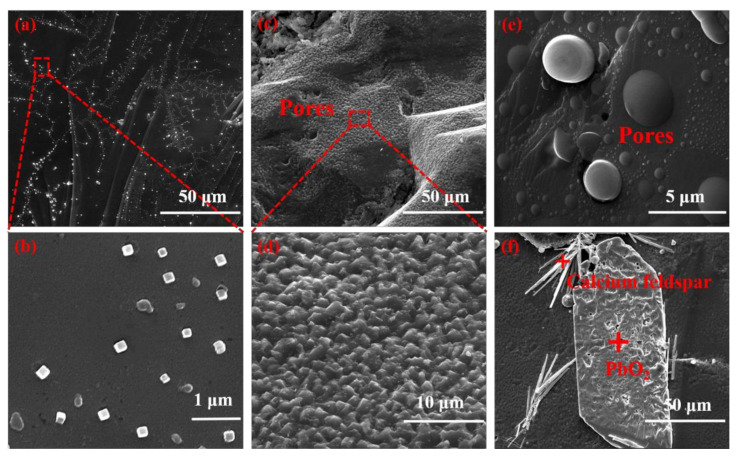
SEM image of pyrite sample. (**a**–**f**) SEM of pyrite samples: the image below is an enlarged version of the red rectangle. The morphology and material distribution on the surface of pyrite can be observed.

**Figure 4 micromachines-12-01555-f004:**
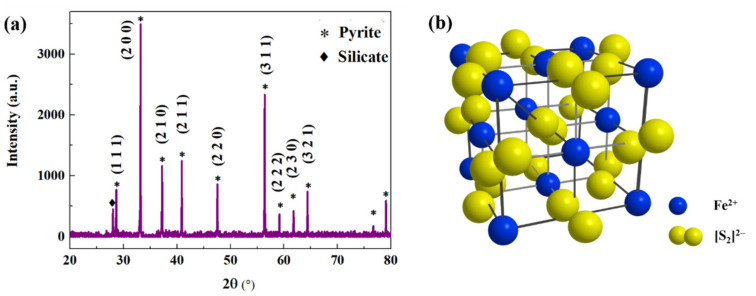
(**a**) The XRD pattern of original pyrite and (**b**) the crystal structure of FeS_2_.

**Figure 5 micromachines-12-01555-f005:**
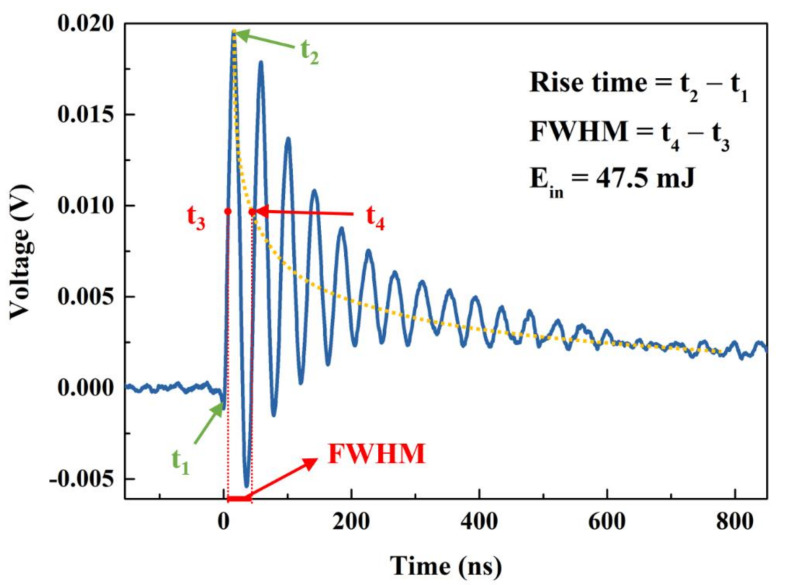
Response of pyrite under the irradiation of 248 nm laser with E_in_ = 47.5 mJ.

**Figure 6 micromachines-12-01555-f006:**
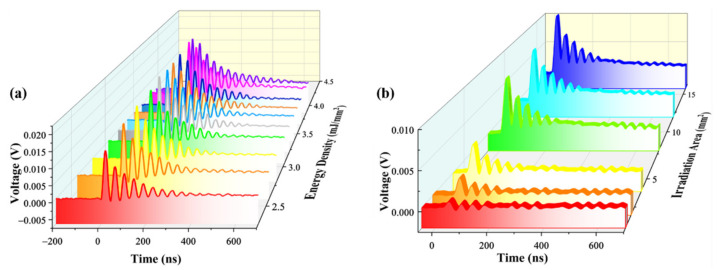
(**a**) LIV signal of pyrite under different laser energy densities from 2.3 to 3.96 mJ/mm^2^. (**b**) LIV signal with laser irradiation area of 1 mm^2^, 2 mm^2^, 4 mm^2^, 8 mm^2^, 12 mm^2^ and 16 mm^2^, respectively.

**Figure 7 micromachines-12-01555-f007:**
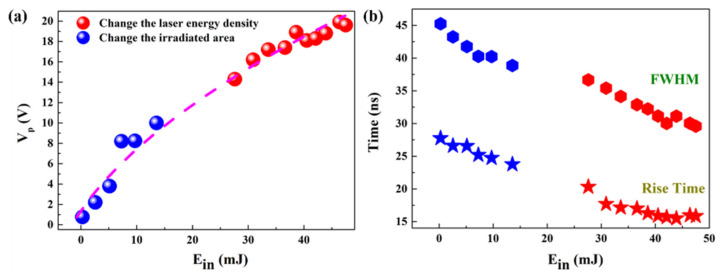
(**a**) V_p_ as a function of the E_in_ of a 248 nm laser. (**b**) Variation of rise time and FWHM with E_in_ of pyrite sample.

**Figure 8 micromachines-12-01555-f008:**
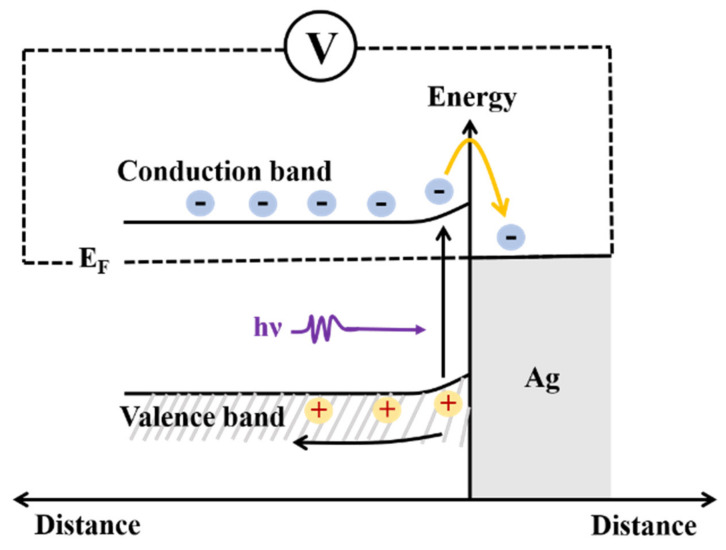
The model of the pyrite interface proposed in this work.

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
