# Peer review of "Ultrafast Deep-Ultraviolet Laser-Induced Voltage Response of Pyrite"

_micromachines, 2021, doi:10.3390/mi12121555_

Round 1

Reviewer 1 Report

The paper as is may not be of an interest to anyone. The detection of UV radiation in short time seems to have no practical application at the present time. The rationale of the work MUST be improved - some other reasoning should be investigated and considered. 

The investigative process seems like an already existent algorithm: use a material - moreover, a mineral from a specific area on the earth and investigate the response to UV radiation. Why? How this came up? Who needs that?

The methodology brings nothing new to the field.

I believe that if properly conceive, the paper may become interesting. However, at the above issue must be addressed in a new paper. Not same paper - a new structure, a new rationale - one that makes the paper interesting and attractive, different methodology.

Reviewer 2 Report

The manuscript "Ultrafast Deep-ultraviolet Laser-induced Voltage Response of Pyrite" by Xuecong Liu, Yudong Li, Haoqiang Wu, Yawen Yu, Honglei Zhan, Xinyang Miao, and Kun Zhao describes the use of pyrite as an optoelectronic, active material. The overall topic is quite interesting and would certainly be suitable for the readership of the journal Micromachines. However, there are a number of concerns that need to be addressed, before publication can be considered.

1. The structure, properties etc. of pyrite should be already defined in the introduction and then a more detailed description should be placed in the materials and methods section. Similarly, the concept of LIV should also be shortly defined and explained in the introduction while the specific working principles for the case of pyrite should be described in even more detail in the subsequent text. As of now, the working principle of LIV is not explained satisfactorily.
2. On page 2, lines 74 - 76, the authors describe the structure and space group of pyrite. They should visualize this via an image of the space group or the unit cell.
3. In Fig 3, the inset shows the use of a MASER. Is this correct or a mistake?
4. On page 5, the authors describe in the text how the laser radiation interacts with the material upon irradition. The different pathways of the excited charge carriers should be visualized via an energy diagram, or something similar. Also, Fig 6 should be used as a basis to clearly explain how the processes of LIV and within the pyrite occur.
5. Currently, the paper lacks quantitative results with regards to the voltage response of the pyrite. As a suggestion, the authors should consider to determine typical parameters used to quantify photodetectors (responsivity, linear dynamic range, detectivity, noise equivalent power, cutoff frequency, etc.). Furthermore, these quantitative results should then be put into context with other systems and literature values.
6. Minor issues regarding spelling, phrasing, and layout can be found in the attached document.

Reviewer 3 Report

The manuscript is reporting the nanofabrication of pyrite as a self-powered UV photodetector room temperature. The introduction is poorly written with serious lack of scientific background regarding this field. In fact, the introduction is only half a page which is not sufficient for a scientific publication. A comprehensive literature report is required to provide some background to potential readers.

In the introduction, some wide band-gap semiconductor materials including ZnO, GaN, TiO2 and SiC were named as potential materials reported in the literature for UV detection application. However, no discussion was provided to the readers regarding the sensing performance of any of these materials. The following manuscript are recommended to be discussed in the introduction section to provide the potential readers with some background understanding about these materials:

  • ZnO: https://doi.org/10.1039/C6NR08425G and https://doi.org/10.1021/acsami.9b19423
  • GaN: https://doi.org/10.1063/1.4978427 and https://doi.org/10.1002/aelm.201700036
  • TiO2: https://doi.org/10.1021/acsami.7b18815

The “Materials and Methods” section of the manuscript does not follow the acceptable reporting style of the scientific manuscript. For example, the following paragraphs are reported in this section, while they do not belong to the “Materials and Methods” section: Pyrite framboids are generally spherical aggregates of submicron-size pyrite crystals. In Figure 1(a), crystals and pyrite framboids with the average size of 0.3 μm are generally diffusely distributed. As shown in Figure 1(b) and (c), the surface structures are various. Pyrite is the most abundant and ubiquitous sulfide mineral in ore deposits, while contains mainly iron, oxygen, sulfur and trace elements such as Pb, Au and Mo [15].

And

“The X-ray diffraction (XRD) patterns of the pyrite sample is displayed in Figure 2, showing the main diffraction peaks of (1 1 1), (2 0 0), (2 1 0), (2 1 1), (2 2 0), (3 1 1), (2 2 2), (2 3 0), and (3 2 1) which were corresponding well to the characteristic peaks of pyrite structure with high crystallinity [16]. The pyrite in the sample has a similar structure with the Pa3 space group. Fe atoms are situated at the face-centered site of one sublattice and S2 molecules are situated on the other, giving a NaCl-like structure [17]. The dominant mineral phases were pyrite, with small amounts of silicate. A pyrite sample was evenly cut into 7 × 4 × 2 mm cubes. Two 2 × 4 mm2 silver electrodes were separated on the polished surface by 2 mm for LIV detection.”

The paragraphs mentioned above are more suitable for the “results and discussion” section rather than “Materials and methods”.

In the “Materials and Methods” section, the authors are expected to describe the materials used in the study, explain how the materials were prepared for the study, describe the research protocol, explain how measurements were made and what calculations were performed, and state which statistical tests were done to analyze the data.

Please revise the manuscript accordingly.

The conclusion is quite simple with no scientific value. A more comprehensive conclusion is recommended.

Author Response

Dear Editors and Reviewers:
Thank you for your letter and for the reviewers’ comments concerning our manuscript entitled “Ultrafast deep-ultraviolet laser-induced voltage response of pyrite” (ID: micromachines-1459016). Those comments are all valuable and very helpful for revising and improving our paper, as well as the important guiding significance to our researches. We have studied comments carefully and have made correction which we hope meet with approval. Revised portion are marked in red in the paper. We appreciate for Editors and Reviewers’ warm work earnestly, and hope that the correction will meet the approval. Please see the attachment for details.
Once again, thank you very much for your comments and suggestions.
Prof. Dr. Kun Zhao

Round 2

Reviewer 1 Report

Congratulations for the accepted paper

Author Response

Dear Reviewer,

Thank you for your recommendation.

Prof. Dr. Kun Zhao

Dr. Xinyang Miao

Reviewer 2 Report

The revised manuscript has improved considerably in comparison to previous versions. However, there are still several points that should be checked, improved, and corrected before publication may take place.

  1. In the introduction the authors discuss the different energy levels relevant to pyrite (e_g* etc.). In a previous version of the manuscript, the authors included an energy diagram (although it was a bit low-resolution and looked like copied from a textbook). Indeed, the manuscript would improve by adding such an energy diagram specifically made to accompany the text in the introduction explaining the different energy levels.
  2. In Fig. 3(b), the authors should make sure that the inserted text (i.e. S2, or Fe) can easily be read and that the text is centered on the spheres depicting the different component of the crystal structure.
  3. The authors should introduce numbered equations for the responsivity and detectivity and wherever appropriate.
  4. Can the authors elaborate how pyrite has a bandgap of 0.95 eV, but is still sensitive to UV? Maybe the authors should also add a UV-Vis spectrum of pyrite.
  5. There are still some spelling errors which the authors should double check (although this is more a process for the pre-publishing).

Author Response

Dear Editors and Reviewers:

Thank you for your letter with regard to our manuscript. We tried our best to improve the manuscript and made changes in the manuscript according to reviewers’ comments. We also highlighted the changes to the manuscript within the document by using the red-colored text.  We appreciate for Editors and Reviewers’  warm work earnestly, and hope that the correction will meet the approval.

Once again, thank you very much for your comments and suggestions.

Prof. Dr. Kun Zhao

Dr. Xinyang Miao

Reviewer 3 Report

The authors addressed all the comments raised by the review. The manuscript could be suitable for publication. 

Author Response

(The authors gave the same response as above.)
